# Gonadal Production and Quality in the Red Sea Urchin *Mesocentrotus franciscanus* Fed with Seaweed *Devaleraea mollis* and *Ulva australis* from a Land-Based Integrated Multi-Trophic Aquaculture (IMTA) System

**DOI:** 10.3390/biology14091294

**Published:** 2025-09-19

**Authors:** Matthew S. Elliott, Yuanzi Huo, Mark Drawbridge

**Affiliations:** Hubbs-Seaworld Research Institute, San Diego, CA 92109, USA; melliott@hswri.org (M.S.E.); yhuo@hswri.org (Y.H.)

**Keywords:** uni ranching, roe, enriched seaweed, GSI

## Abstract

Sea urchins are a prized seafood delicacy in many parts of the world, including California, USA and Japan. However, some wild sea urchins have small or poor-quality gonads due to limited food in barren habitats, making them unmarketable despite high demand. This study explored whether feeding red sea urchins with seaweed grown on land in fish farming systems could improve gonad quality. The seaweeds were cultivated in tanks using nutrient-rich water from fish culture, which helps the seaweed grow and become more nutritious. Over an eight-week period, sea urchins were fed one of two types of these enriched seaweeds. The results showed that the urchins developed larger and more colorful gonads, making them more appealing for seafood markets. These findings suggest that using seaweed from fish farms not only improves sea urchin quality but also helps reduce waste from aquaculture. This approach offers a sustainable and efficient way to produce high-quality seafood while benefiting both farmers and the environment.

## 1. Introduction

Over the past four decades, rocky reef ecosystems worldwide have undergone significant shifts, with sea urchin overgrazing driving the widespread formation of urchin barrens, i.e., habitats characterized by high urchin densities and minimal macroalgal cover [1]. These barrens represent a phase shift from productive, kelp-dominated forests to structurally and functionally simplified reef systems that can persist for decades [1,2]. Such ‘phase shifts’ refer to large-scale ecosystem transitions, often from kelp forests to urchin barrens. While sometimes reversible if grazing pressure is reduced, these barrens can also represent stable ecological regimes that persist for decades [1,2]. In regions such as Tasmania, other parts of Australia, and the North Atlantic, these transitions have been driven primarily by unchecked sea urchin grazing following the depletion of key predators [2,3]. In California, USA, particularly along the northern coast, kelp forests have declined by more than 90% since 2014. The decline of nearshore kelp forests has been closely linked to dramatic increases in sea urchin populations, which have intensified grazing pressure and severely limited kelp regrowth [4,5].

Sea urchin densities in California, USA, increased from less than 2 individuals m^−2^ before 2014 to as many as 12.9 individuals m^−2^ by 2015 [4]. The resulting urchin barrens lack sufficient macroalgal food to sustain the urchin populations, leading to metabolic depression in many individuals [5]. In this state, sea urchins reduce energy allocation to reproduction, resulting in small, pale, and low-quality gonads with no commercial value [6]. These persistent, food-limited conditions are a primary factor in the collapse of multiple urchin fisheries, e.g., the red sea urchin (*Mesocentrotus franciscanus*, RSU) fishery in California [4,7]. Similar dynamics and management challenges have been reported for other fisheries such as the green sea urchin (*Strongylocentrotus droebachiensis*) [8].

Historically, the harvesting of several urchin species led to high-value fisheries worldwide [9]. Urchins are mainly harvested for their gonads, often referred to incorrectly as ‘roe’, are highly marketable when the quality is high (e.g., $108.55 per 100 g) [10]. RSUs are the largest sea urchin in the class Echinoidea, which can be found from the Gulf of Alaska near Kodiak Island, Alaska to Baja California, Mexico [11]. RSUs are the basis for an important commercial fishery and serve as an indigenous traditional food resource as well as model species in developmental research [12]. In recent years, urchin barrens have resulted in a collapse of the RSU commercial fishery due to poor gonad production and associated lack of marketability [4]. From 1980 to 2023, RSU fishery landings have decreased from approximately 10,000 to 1100 metric tons in California, USA [7]. Comparable pressures and management challenges have also been observed in *S. droebachiensis*, an important commercial species in the Northwest Atlantic and Barents Sea [8].

Although urchin fisheries have significantly decreased, aquaculture techniques have shown to be a viable means to improve market value of urchins [13,14]. Urchin ranching involves capturing wild urchins and holding them in tanks or cages where they are fed to fatten their gonads to an acceptable market size and quality [15]. Gonad production and quality depend on food quantity, quality, and the rate of consumption, digestion, and absorption [16,17]. Recently, improvements in formulated feeds, freshly collected wild seaweeds, and farmed seaweeds have been shown to improve gonad production and quality of several sea urchin species [18,19,20,21,22,23,24].

The nutritional quality of seaweeds can be significantly improved when co-cultured with other trophic levels in integrated multi-trophic aquaculture (IMTA) systems [25,26,27]. The nutrient-enriched seaweeds from IMTA systems can then be fed to sea urchins collected off barrens to improve their gonad size and quality [28,29]. Beyond direct applications to sea urchin enhancement, land-based IMTA systems also provide broader ecosystem benefits such as filtering seawater and removing trace metals [28]. Green macroalgae *Ulva* spp. has been identified as an ideal biofilter due to its strong capacity for nitrogen and phosphorus uptake and robust resilience to environmental changes [26,30], as well as its wide application in human food, animal feed, biofuel, and medicine [28,29,31,32]. Red macroalgae *Devaleraea mollis* has also been used as a biofilter and feed in abalone polyculture, as well as for human consumption [33,34,35]. *Ulva* and *Devaleraea* species have already been used as feeds to improve production and quality of gonads in barren sea urchins either as whole fresh thallus or as one of the ingredients in formulated diets [21,22,28,36,37]. The fast-growing and high protein content demonstrated by *Ulva* and *Devaleraea* in IMTA systems make them good choices as feed options for urchin gonad development.

Until now, no studies have been published on the gonadal size and quality improvements of RSUs from barrens fed either formulated diets or seaweeds. In this study, *U. australis* and *D. mollis* were cultured in the effluent from white seabass (*Atractoscion nobilis*, WSB) culture tanks in an IMTA system and then fed to starved RSUs to improve the size and quality of gonads. The aims of this study were to measure the feed intake of IMTA-produced *U. australis* and *D. mollis* by starved RSUs; to determine whether these two nutrient-enriched seaweed species could significantly improve the size and quality of gonads; and over what time period; and also to evaluate whether RSUs can be incorporated into IMTA systems to consume produced nutrient-enrichment seaweeds. We hypothesized that (1) RSUs would efficiently consume IMTA-produced *U. australis* and *D. mollis*; (2) both species would significantly increase gonadal size; (3) gonad quality would improve to market-acceptable grades over time; and (4) RSUs can be incorporated into IMTA systems to diversify seafood production.

## 2. Materials and Methods

### 2.1. Red Sea Urchin Sourcing

Red sea urchins (RSUs) (n = 40) were collected from Point Loma, San Diego, California, USA in January of 2023 at a subtidal (5–10 m deep) barren by a commercial fisherman using SCUBA. The barren extended across approximately 0.5 km^2^, characterized as a rocky substrate with little surrounding kelp. The RSUs were then transported within 20 min to Hubbs-SeaWorld Research Institute (HSWRI, San Diego, CA, USA) in coolers filled with only ambient sand filtered seawater (15–16 °C) where phenotypic measurements were taken (mean diameter = 104.92 ± 13.67 mm, mean height = 61.45 ± 10.59 mm). At HSWRI, RSUs were placed into a 1500 L acclimation tank on flow through, UV-sterilized seawater with an exchange rate of 72 times per day. Ambient seawater temperature was 14–16 °C and the RSUs were held there for 7 days until the start of the experiment. The acclimation tank was covered with 90% shade to control light exposure. During the acclimation period, RSUs were not fed, and feces were siphoned out daily.

### 2.2. Seaweed Production Methods

*Devaleraea mollis* and *U. australis* were cultured in a flow through IMTA system co-cultured with WSB, using sand-filtered seawater. The *Ulva* used in this study was initially identified morphologically as *U. lactuca* at the time of collection, but subsequent DNA barcoding confirmed it as *U. australis*. The IMTA system was constructed in four step-down tiers that each supported a circular black polyethylene tank of 700 L (height: 70 cm; surface diameter: 60 cm; bottom diameter: 52 cm) and a surface area of 1.09 m^2^. The tiered tanks allowed seawater to cascade by gravity from the first to the last prior to discharge such that water was only pumped once to the system. Compared with other IMTA designs, cascade systems provide a more cost-effective way to distribute seawater across different trophic-level tanks without additional pumps, while still maintaining replication and control. Each tiered unit of four tanks was set up in triplicate rows for a total of 12 tanks [26]. WSB were stocked in the first tier of tanks at a density of 30 kg m^−3^. WSB were fed a commercial diet (Skretting 4.0 mm Marine mix with a proximate composition of 46% protein and 12% fat) to satiation at a feeding rate of 1–3% body weight daily depending on water temperature. The seawater exchange rate cascading through each tank was set to 63 vol. day^−1^ according to our previous research [25]. These two seaweed species were also cultivated in tanks with raw seawater for comparing the biochemical composition with those in the IMTA system. The initial stocking density of *U. australis* and *D. mollis* was the same for both seawater treatments at 1 kg m^−2^ and 4 kg m^−2^, respectively. In tumble culture, both *U. australis* and *D. mollis* were vegetatively propagated, and biomass was weighed and reset to the initial density weekly. Fresh *U. australis* and *D. mollis* biomass was harvested directly from the IMTA tanks and fed to experimental RSUs three times per week. Sub-samples (200 g) of seaweed were collected for measurement of biochemical composition at the beginning, weekly and end of the trial from each tank. Tissue nitrogen and carbon of *U. australis* and *D. mollis* were measured using a Costech 4010 gas chromatography elemental analyzer (Costech Analytical, Valencia, CA, USA). The protein content of *U. australis* and *D. mollis* was determined by multiplying the nitrogen content by 6.25 [38].

### 2.3. Feeding Trials

RSUs were divided evenly into two fiberglass circular white tanks (n = 20 RSUs per tank), each with a working volume of 175 L. Each of the tanks was plumbed with a central 3.80 cm diameter standpipe at a height of 30 cm with an aeration ring at the bottom of the standpipe. Incoming ambient seawater was pumped from the adjacent lagoon through sand filters into a holding tank that allowed for degassing and also ensured uninterrupted water flow. Seawater then gravity fed into the culture tanks, and effluent was discharged back to the lagoon. Each tank also had a 90% shade cloth over the top to create a darker, more natural lighting environment. Seawater flow rate was set to 7.60 L per min (L min^−1^) which equated to 63 exchanges per day. The mean diameter and height of RSUs in each tank corresponded to a volume of 785.40 cm^3^ per individual urchin. Total RSU volume per tank was 15.70 L, representing less than 10% the total tank volume. This low density ensured sufficient space and oxygenation for all individuals. Importantly, feeding activity was not limited at this density; observations confirmed unrestricted access to feed, aided by seaweed tumbling created by the aeration ring. These stable and uniform conditions minimized competitive interactions, reduced stress, and maintained consistent environmental parameters across individuals. Additionally, low within-tank variation in body size and behavior meant that each urchin could be treated as an independent replicate. Given these conditions, each RSU functioned effectively as an independent replicate, and the use of five individuals per treatment group (n = 5) was deemed appropriate for assessing dietary effects. Since the focus was on assessing the feasibility of incorporating RSUs into an IMTA framework, the feeding trials were based exclusively on seaweeds produced within the IMTA system. Raw seawater-fed seaweed was not included as feed, as the aim was to evaluate performance when provided with nutrient-enriched seaweed generated by fish effluent, the intended operational scenario for such a system. After a one-week acclimation period in the experimental tanks, the RSUs were fed with either *U. australis* or *D. mollis* from the WSB IMTA at a daily ration of 2–3% of total body wet weight, calculated based on the average body weight of all RSUs within each tank. A 2–3% daily ration was selected based on prior studies showing optimal gonad development [15,17,39]. Fresh seaweed was added every other day at 9:00 am, and uneaten seaweed and feces were removed and weighed for measurements. The wet weight (WW) and dry weight (DW) of uneaten seaweed and feces were measured in order to calculate ingestion rate (IR), fecal rate (FR), absorption efficiency (AE). WW of seaweed was measured by placing seaweed from the culture tank onto a net for 5 min, then weighing it to the nearest 1.00 g. DW was measured after the seaweed was fully dried at 60 °C for 48 h. Since GSI can be influenced by body size, RSU diameter and height were used to assess size differences between treatments. To account for progressive reductions in tank biomass as RSUs were removed for sampling, feeding rations were recalculated every four weeks based on the remaining urchins’ average body weight. RSU body weight was measured at each sampling point to track biomass changes over time. The experiment lasted for 56 days. At weeks four and eight, five RSUs per treatment were sacrificed for analysis of gonadosomatic index (GSI), and gonad quality [40].

### 2.4. Environmental Factors and Water Quality

The temperature, pH, and dissolved oxygen (DO) were measured using a Hach HQ40d handheld multimeter (Hach, Loveland, CO, USA), TGP probe (OxyGuard, Farum, Denmark), and Pinpoint pH meter (American Marine, Ridgefield, CT, USA) twice daily at 8:00 am and 16:00 pm. Additionally, total ammonia nitrogen (TAN) and phosphate (PO_4_–P) were measured weekly using a Hach model DR-6000 (Hach, USA). Measurements were taken from the influent and effluent of each culture tank.

### 2.5. Measuring Protocol

Ingestion rate (IR, g DW ind.^−1^ day^−1^) was calculated as the difference between the fed seaweed biomass and the uneaten biomass in g of DW per RSU daily:IR = (Amount offered (g) − amount uneaten (g))/Time (days)/Number of RSUs(1)

Fecal rate (FR, g DW ind.^−1^ day^−1^) was calculated as the amount of feces in g DW produced per RSU daily:FR = (Amount of feces (g)/Number of RSUs/Time (days).(2)

The absorption efficiency (AE, %) was calculated using the amount of feed eaten and the amount of feces produced per RSU daily:AE = 100 × (amount eaten (g) − the amount of feces (g))/(amount eaten (g)).(3)

GSI (%) was expressed as the percentage of gonad wet weight to total body wet weight:GSI = 100 × (gonad wet weight (g)/whole urchin wet weight (g)).(4)

Additionally, weekly GSI_increase_ (% wk^−1^) and Gonad wet weight_increase_ (g wk^−1^) was calculated as:GSI_increase_ = (GSI_end_ − GSI_initial_)/Weeks.(5)Gonad wet weight_increase_ = (gonad wet weight_end_ − gonad wet weight_initial_)/Weeks.(6)

### 2.6. Gonad Quality

RSU gonad quality and grade were assessed using a modified protocol based on [40], as shown in Table 1. The grading levels for gonad color were further refined based on red, green, and blue (RGB) values (Table 2). Gonad color was measured using a photo-box (Glendan light box, USA, 5500 K, CRI ≥ 95 LEDs) with a color card (Datacolor SpyderCHECKR 24, Jiangsu, China) and images were taken manually using an iPhone 15 camera (Apple, Cupertino, CA, USA) set at standard photo settings, placed 30 cm on top of the photo-box through the viewing port. Digital photographs of gonads were then analyzed using Adobe Photoshop (PS; version 24.7; Adobe Inc., San Jose, CA, USA). Using the color-picker in the PS software, five points were randomly sampled for color on each gonadal lobe image from each individual RSU. The five RGB values obtained by PS were averaged for each individual RSU. These averaged RGB values were compared to an updated color grading scale developed for this study (Table 2). The scale classified RSU gonad color into four levels: Level I: Bright yellow-orange (R: 160–255, G: 115–255, B: 0–125), Level II: Dull yellow-orange to light orange-brown (R: 120–160, G: 75–125, B: 0–95), Level III: Light to moderate brown (R: 100–120, G: 50–115, B: 0–65), and Level IV: Dark brown or black (R: 0–100, G: 0–80, B: 0–35). To measure texture, image analyses software (Leica LAS-X, version 4.13), was used with a dissecting scope at 4× magnification. A digital image of each gonadal lobe (n = 5) was taken using imaging software, then measuring lines were drawn across 10 individual granulations per gonad to give 10 measurements to the nearest 0.01 mm. The size of each granulation was then averaged per gonad lobe, then per individual, to give a texture score. Gonad granulation sizes greater than 1.00 mm were graded as fine (F), 1.00–2.00 mm as medium (M), and less than 2.00 mm as course (C). To measure firmness, a small balance weight was placed on top of each gonad lobe (72 g or 175 g) following the methods of [40]. If gonads remained whole during dissection and with 72 g on top of the gonad, they were deemed firm. If gonads did not remain whole during dissection but held up at 175 g, then they were also deemed firm [40].

### 2.7. Data Analysis

Data were expressed as mean ± standard deviation. Data analysis was performed by R software (version 4.2.2). Tests of homogeneity of variance were assessed using the Shapiro–Wilk and Levene’s tests, respectively. Percentage data were arcsine-transformed for normalization before analysis. One-way ANOVA was used to assess differences in seaweed chemical composition between IMTA-cultured and raw seawater-cultured treatments. Two-sample *t*-tests were used to compare IR, FR, and AE of RSUs fed with different diets, as well as GSI, gonad wet weight, and their weekly changes. Additional *t*-tests were conducted to compare morphometric parameters between dietary treatments at both weeks 4 and 8. Differences were considered significant at *p* < 0.05.

## 3. Results

### 3.1. Environmental Factors and Water Quality

The temperature, DO and pH ranged from 13.58 to 19.36 °C, 7.25 to 9.03 mg L^−1^ and 7.84 to 8.22, respectively, throughout the experimental period. The average TAN and PO_4_-P concentration of influent in seaweed tanks (equal to effluent of WSB tanks) in the IMTA system was 0.09 ± 0.05 mg L^−1^ and 0.72 ± 1.04 mg L^−1^, respectively. This was higher than those in raw seawater, with average concentrations of 0.02 ± 0.03 mg L^−1^ and 0.22 ± 0.21 mg L^−1^, respectively. The TAN and PO_4_^−^ concentrations in the IMTA system were both significantly higher than the TAN and PO_4_^−^ concentrations in the raw seawater (*p* < 0.01 and *p* = 0.02), respectively.

### 3.2. Chemical Composition of U. australis and D. mollis

The average nitrogen content of *U. australis* and *D. mollis* cultured in WSB effluent was 3.48 ± 0.26% dry weight (DW) and 4.88 ± 0.32% DW, and the average carbon content was 28.83 ± 2.00% DW and 30.78 ± 1.61% DW, respectively (Figure 1). The nitrogen and carbon content of *U. australis* cultured in the IMTA system was significantly higher than *U. australis* cultured with raw seawater (*p* < 0.01 and *p* = 0.03). For *D. mollis*, the nitrogen content was significantly higher when cultured in the IMTA tanks than cultured in the raw seawater tanks (*p* < 0.01), but the carbon content did not show a significant difference (*p* = 0.38). Based on the nitrogen content, the average protein content of *U. australis* and *D. mollis* was 21.74% DW and 30.51% DW when cultured in the IMTA tanks, respectively.

### 3.3. Red Sea Urchin Height, Diameter, Ingestion Rate, Fecal Rate, and Absorption Efficiency

The beginning height and diameter of red sea urchins (RSUs) in the *D. mollis* treatment was similar to RSUs in the *U. australis* treatment (*D. mollis*: 62.83 ± 9.75 mm and 101.78 ± 10.11 mm, *U. australis*: 58.85 ± 5.82 mm and 106.11 ± 18.36 mm). There were no significant differences in RSU diameter or height between dietary treatments at the beginning of the experiment (*p* = 0.14 and *p =* 0.84). The ingestion rate (IR) and fecal rate (FR) of RSUs fed with *D. mollis* were all higher compared to RSUs fed with *U. australis*, with average values of 1.71 ± 0.63 g ind.^−1^ day^−1^ and 1.25 ± 0.87 g ind.^−1^ day^−1^ for IR, and 0.24 ± 0.28 g ind.^−1^ day^−1^ and 0.01 ± 0.01 g ind.^−1^ day^−1^ for FR, respectively (Figure 2A,B). Conversely, the absorption efficiency (AE) of RSUs fed with *U. australis* was significantly higher than RSUs fed with *D. mollis* ranging from 99.44 to 99.98% and 96.27 to 99.83%, respectively (*p* = 0.03, Figure 2C).

### 3.4. Gonadal Somatic Index (GSI) and Gonad Wet Weight

RSUs fed *D. mollis* showed higher GSI values and gonad wet weight (WW) at weeks 4 and 8 (6.05 ± 2.38% and 6.35 ± 2.60%, and 27.56 ± 18.90 g and 31.20 ± 7.20 g, respectively, (Figure 3A,B) compared to RSUs fed *U. australis* (4.02 ± 1.04% and 4.64 ± 1.48%, and 14.74 ± 7.20 and 23.04 ± 10.20 g, respectively). Additionally, the weekly increase in GSI (GSI*_increase_*) and gonad WW in RSUs fed *D. mollis* was higher than RSUs fed *U. australis*, with average values of 0.37 ± 0.29% wk^−1^ and 2.77 ± 0.89% wk^−1^, and 0.18 ± 0.16% wk^−1^ and 1.75 ± 1.28% wk^−1^, respectively (Figure 3C,D). There was no significant difference in RSU GSI or gonad WW between those fed *D. mollis* and *U. australis* throughout the experiment (*p* = 0.20 and *p* = 0.18, respectively).

### 3.5. Gonad Quality

At the start of the trial, gonad color levels for RSUs were classified as Level IV or Level III (Table 3A). Over the 4-week and 8-week feeding period, the gonad color gradually improved to Level II for all RSUs fed *U. australis* (Table 3B,D), and four out of five RSUs fed *D. mollis* (Table 3C,E). For texture, RSUs were classified as fine or medium, with average values of 1.06 ± 0.24 mm at the start of the trial (Table 3A). By the end of the 8-week feeding period, texture improved, with average values decreasing to 0.84 ± 0.15 mm for RSUs fed *U. australis* (Table 3D) and 0.99 ± 0.11 mm for RSUs fed *D. mollis* (Table 3E), respectively. For firmness, three out of five RSUs were classified as ‘F’ at the start of the trial (Table 3A). By the end of the 8-week feeding period, all RSUs fed *U. australis* were classified ‘F’ (Table 3D), and four out of five RSUs fed *D. mollis* also achieved ‘F’ (Table 3E). The overall grade of RSUs was initially C or D (Table 3A). After the 4-week feeding period, grades significantly improved for both treatments (Table 3B,C). By the end of the 8-week feeding period, all RSUs fed *U. australis* were graded as B (Table 3D), while four of five RSUs fed *D. mollis* also achieved a grade of B (Table 3E).

## 4. Discussion

Nutritional quality of *D. mollis* and *U. australis* was improved when cultured in the nutrient-enriched effluents from WSB tanks in this flow through seawater IMTA system compared with those cultured in raw seawater, which was also verified in our previous studies [25,26,41]. Data from another study we conducted (Unpublished work, Elliott et al.) indicated that IMTA-cultured *D. mollis* and *U. australis* contained as high as 41.40% and 28.10% protein, respectively, with comparable ash content (21.50%) and modest differences in fat (2.98% vs. 4.25%) and fiber (3.34% vs. 4.56%) content. In a nutrient profiling study, *D. mollis* and *U. lactuca* grown with WSB effluent contained 4.89% and 3.48% nitrogen, respectively, corresponding to protein values of approximately 30.6% and 21.8% [26]. Additionally, a gonad enhancement study using another dulse species, *Palmaria palmata* (41% protein), cultured in enriched laboratory conditions, improved gonad coloration in *Psammechinus miliaris* [18], suggesting that dulse has benefits as a diet for other echinoid species. Moreover, these traits highlight why *Ulva* and *Devaleraea* are particularly suitable for RSU cultivation; both species combine high protein levels and proven efficiency as biofilters in IMTA systems.

For *Ulva* spp., multiple IMTA studies support their high nutritional value and consistency. In a one-year study of IMTA-cultured *U. lactuca*, protein levels ranged from 25.2 to 28.4% and remained stable across seasons, with low lipid and moderate fiber content [42]. In a separate gonad enhancement study, *U. lactuca* cultured in fish effluent reached 35.7 ± 1.1% protein and significantly improved gonad quality in *Paracentrotus lividus*, with GSI reaching 19.6 ± 3.75% and more than 75% of urchins producing Grade A gonads [28]. These values support the application of *U. australis* in high-quality feed formulations for enhancing sea urchin gonad production.

Although high-protein seaweeds are often associated with improved gonad production, responses vary by species. In a feeding trial, *Mesocentrotus nudus* fed *Pyropia yezoensis* (36.7% protein) exhibited reduced gonad quality [43]. *Tripneustes gratilla* fed *Ulva* spp. increased GSI by 64% after 12 weeks [44]. Lastly, *Strongylocentrotus purpuratus* reached market size GSI after 10 weeks when fed *Gracilaria pacifica* [14]. Similarly, *P. miliaris* fed *D. mollis* (41% protein) reached a GSI of 11.4%, compared to 7.5% in *P. lividus* [17]. These outcomes underscore the importance of aligning diet composition with species-specific physiological needs.

In this study, red sea urchins (RSUs) showed relatively high ingestion rates (IRs) when fed IMTA-produced fresh *D. mollis* and *U. australis*, which surpassed the consumption rates reported for other species of sea urchin, likely due to the RSUs’ larger body size (up to 635.20 g body wet weight and 176.65 mm in diameter). Given their larger body mass, RSUs require greater food intake to sustain their basal metabolic functions and restore their gonadal tissues. This is similar to findings with other sea urchins like *M. nudus* (weighing 39.80 g to 45.80 g) when fed nutrient-enriched *P. yezoensis* [43], *Lytechinus variegatus* (measuring 5 to 8 cm) when consuming six different seaweed species [45], and *S. purpuratus* (averaging 30 ± 2.80 g) when fed kelp *Macrocystis pyrifera* and *U. lactuca* enriched with nutrients [46]. Additionally, unlike smaller urchin species, RSUs exhibit considerable variability in wet body weight due to water retention. This variability makes total wet weight a poor metric for phenotyping. Instead, this study used test diameter and height to compare morphometrics between groups. No significant differences in diameter or height were observed between treatments at weeks 0, 4, or 8 (*p* > 0.05), validating the experimental design used for RSUs. Conversely, RSUs had very low fecal rates (FRs) when fed IMTA-produced fresh *D. mollis* and *U. australis* in this study, with average values of only 0.24 ± 0.28 g ind.^−1^ day^−1^ and 0.01 ± 0.01 g ind.^−1^ day^−1^, respectively. The FRs of other sea urchins fed seaweeds were not directly reported, but values of FR can be inferred by their digestibility values. High IRs and concurrently low FRs would result in high digestibility, which is indicated by the high absorption efficiency (AE) of 99.44–99.98% and 96.27–99.83% for RSUs fed *U. australis* and *D. mollis* in this study. Similar results were also obtained by others, e.g., *M. nudus* had a digestibility of 96.20% when fed a high-protein concentrated *P. yezoensis* [43]; *T. gratilla elatensis* had a digestibility of 81.4% when fed IMTA-produced *U. lactuca* [28], all of which reported relatively low FRs when fed nutrient-enriched seaweeds. According to the relatively high IRs, low FRs and calculated high AEs in this study, it can be concluded that the IMTA-produced *U. australis* and *D. mollis* were very palatable to the RSUs.

The GSI of RSUs fed with IMTA-produced *U. australis* and *D. mollis* increased from 3.00% at the start to 4.64% and 6.35%, respectively, over the 8-week feeding period in this study. These relatively low values and small increase make it challenging to compare results across different sea urchin species due to varying body sizes, weights, and experimental durations. Typically, smaller sea urchin species, which have shorter lifespans, reach higher GSIs over shorter feeding periods. For example, the GSI of *S. nudus* (50 mm) increased from 6.50 to 18% within two months when fed with *Laminaria religiosa* [47], and *S. purpuratus* (52.50 ± 8.20 mm) reached up to 22.40 ± 1.68% in nine weeks [13]. Smaller sea urchin species generally have higher GSIs compared to larger species because their gonads constitute a greater proportion of total body mass. This pattern reflects allometric scaling, where larger species allocate proportionally more biomass to somatic tissues than to reproductive organs [48]. In the case of RSUs, which have a minimum market size of approximately 83 mm in diameter in Southern California [12], GSIs from wild collections were reported between 9.86 and 11.16% during transport [49]. Stefánsson et al. [10] reported a potential GSI of up to 20% during peak reproductive periods, but this figure was calculated using “drained body weight”. If calculated based on “total live body weight”, the GSI would be significantly lower than 20% because sea urchins contain substantial volumes of perivisceral fluid, often comprising 15–25% of total body mass [50]. Moreover, GSI of RSUs typically ranged between 1 and 5% during non-reproductive periods. Thus, RSUs likely require more time to achieve higher GSIs compared to smaller species. Extrapolating from this data, the GSI of RSUs fed *D. mollis* increased much more rapidly from the start of the trial to week 4 compared to those fed *U. australis*. This rapid increase in the *D. mollis* group could be attributed to the higher nitrogen and protein content of *D. mollis*, which has been shown to support faster gonadal development in sea urchins [18]. Additionally, red algae like *D. mollis* have more favorable amino acid and carotenoid profiles suitable for digestion [38]. The large error ranges observed in some treatments likely reflect natural variability in RSU responses, consistent with their wide size range and variable reproductive condition at collection. In contrast, the *U. australis* group exhibited a more linear growth trend in GSI throughout the experiment, consistent with findings that seaweeds with moderate protein content can sustain consistent gonadal growth over time [46]. Based on the observed linear growth in the *U. australis* treatment, it can be estimated, assuming no seasonal or reproductive demands, that RSUs would require approximately 12–14 weeks to reach a GSI up to 10% when fed with IMTA-produced *U. australis* and *D. mollis*, a hypothesis that will be tested in future studies. While this extrapolation requires further validation, these findings suggest potential scalability. In California, USA and other regions where seaweed aquaculture is expanding, IMTA systems could be adapted at farm scale to enhance uni, though feasibility will require additional evaluation. IMTA systems may also provide added value through water-quality improvements and contaminant reduction, offering broader benefits for aquaculture sustainability and ecosystem health [28].

In this study, the quality of RSU gonads was evaluated using indices of red, green, and blue (RGB) color analysis for gonad color, as well as texture and firmness [39], which were then combined for each RSU into a grade. Notably, this is the first published study to establish a standardized RGB-based color grading scale for sea urchin gonads. Previous studies assessing gonad quality have generally relied on subjective visual assessments or categorical descriptors such as “bright orange”, “dull yellow”, or “pale brown” [28,29,36,39,43]. While these qualitative approaches have been useful in industry settings, they lack consistency, are subject to evaluator bias, and are difficult to replicate across laboratories, markets, and geographic regions. Other methods, such as spectrophotometry or image-based color clustering, have been explored in seafood quality analysis (e.g., tuna and salmon flesh) but remain largely impractical for sea urchin grading due to equipment cost, complexity, and the necessity of highly controlled lighting environments [51].

To address these limitations in traditional sea urchin gonad assessment, we developed a method that quantifies gonad color using RGB values extracted from standardized digital photographs. Gonads were photographed under consistent lighting with a neutral white background, and RGB values were extracted from multiple regions per gonad using Adobe Photoshop’s color sampling tool. These values were averaged across all five lobes of each individual and converted into a color grade using an empirically derived scale. This scale was based primarily on the red (R) channel, which closely correlates with perceived “freshness” and quality in echinoderm gonads, while green (G) and blue (B) values were used to confirm color balance and identify outliers such as browning or bleaching.

The grading scale was developed from hundreds of measurements and calibrated using natural breaks in RGB distributions across individuals. Grades were not arbitrarily spaced but instead determined by analyzing clusters of R values and matching them with perceptual color shifts visible in the gonad images. Importantly, within the RGB color model, relatively small changes in one channel, particularly red, can cause a disproportionate shift in the final perceived color, especially near transition thresholds. For instance, when R drops below a certain intensity while G or B values remain elevated, the color can abruptly shift from vibrant orange to dull brown or gray. These tipping points helped define the upper and lower bounds for each color grade, ensuring that transitions between categories reflected meaningful changes in visual quality rather than minor numeric differences.

The main advantage of this method lies in its simplicity, objectivity, and scalability. It requires only a smartphone camera, widely available photo-editing software, and a standardized photographic set up, making it accessible to most aquaculture operations. The RGB system also has potential for integration into automated image analysis workflows, enabling high-throughput, real-time quality control in commercial uni production. While this represents a novel advancement, the method also has limitations that should be considered. Even though RGB values effectively differentiate color, they do not directly quantify pigment composition, such as carotenoid or astaxanthin content. Results are also sensitive to lighting conditions, and consistent image acquisition protocols are necessary to ensure comparability across datasets. Future improvements should focus on pairing RGB scoring with pigment assays, integrating calibrated lighting environments, and potentially expanding the system into perceptually uniform color spaces such as CIELAB (L*a*b*), which more closely align with human visual perception. Broader adoption will also benefit from cross-laboratory validation, the development of standardized reference color libraries, and open-access grading tools to promote consistency in urchin quality research and marketing.

The grades of the gonads from this study showed significant improvement at the end of the trial compared to the initial quality when fed either *U. australis* or *D. mollis*. The gonad grade of RSUs fed with *D. mollis* was better than those fed with *U. australis* after four weeks but was similar in both experimental groups after eight weeks (Table 3). Gonad color is one of the most important parameters for determining sea urchin gonad quality [36,39], and it was the main contributor to the improved quality of RSU gonads fed with *U. australis* and *D. mollis* in this study. Echinenone, synthesized from β-carotene, is responsible for the yellowish-orange color of high-quality sea urchin gonads [52]. It has been reported that IMTA-produced *U. lactuca* constitutes a potential source of pigments such as β-carotene, lutein, and violaxanthin [53]. Total carotenoid content in red macroalgae is reported to be approximately 800 µg g^−1^ dry weight (DW), including 30 µg g^−1^ β-carotene [54]. A formulated diet containing *D. mollis* significantly changed the shell color of Pacific abalone (*Haliotis discus hannai*) resulting in the development of market-preferred dark-brown shells compared to the lighter yellow or pink shells observed in abalone fed artificial diets [55]. The improvement in RSU gonad color could likely be attributed to the IMTA-produced *U. australis* and *D. mollis*, with IMTA conditions known to elevate carotenoid levels [53]. However, specific β-carotene levels in these two seaweeds should be measured in future studies. The texture index between the two feeding groups was similar and comparable to the initial texture values, while the gonad firmness of RSUs fed either *U. australis* or *D. mollis* improved over the eight-week study. This may be due to changes in moisture content in the RSU gonads by the end of the experiment [56], which will be evaluated in future studies. Gonad taste is also a very important factor in the commercial uni industry [39]. Taste evaluation can be performed by having multiple experienced tasters score the samples after tasting [36,39]. Unfortunately, in this study, there were not enough gonad samples to set up formal taste tests, but a few samples were sent to a local distributor to test whether the taste was the same or improved compared to freshly caught RSUs. Feedback indicated that the taste of RSU gonads fed IMTA-produced *U. australis* and *D. mollis* was comparable to that of wild-caught RSUs (Dave Rudie, personal communication). Additionally, the sweet amino acid glycine and bitter amino acids valine and lysine can be measured to evaluate gonad taste [43]. Detailed research on taste evaluation will be conducted in future studies.

## 5. Conclusions

Red sea urchins (RSUs) represent a very important commercial fishery along the Pacific coast; however, environmental changes including declining kelp forest populations, are contributing to reduced gonad production and localized fishery declines [57]. This study demonstrated that the nutritional quality of *U. lactuca* and *D. mollis* can be enhanced by cultivating them in the effluent of WSB culture tanks in an IMTA system. RSUs collected from a barren consumed these two seaweed species readily, and improvements were observed in their GSI and overall gonad quality over an 8-week feeding period. Extending the feeding period would likely continue to yield further enhancements in GSI and gonad quality. Although this study did not test formulated diets or other seaweed species due to limited animal availability, these aspects will be explored in future research. The findings also underscore the potential for integrating RSUs into land-based IMTA systems (incorporating WSB, RSU, and seaweeds) to diversify seafood production and enhance aquaculture system efficiency. Importantly, this study introduces the first standardized red, green, and blue (RGB)-based gonad color grading scale for RSUs, providing a replicable and objective tool to evaluate and improve gonad quality in aquaculture settings.

## Figures and Tables

**Figure 1 biology-14-01294-f001:**
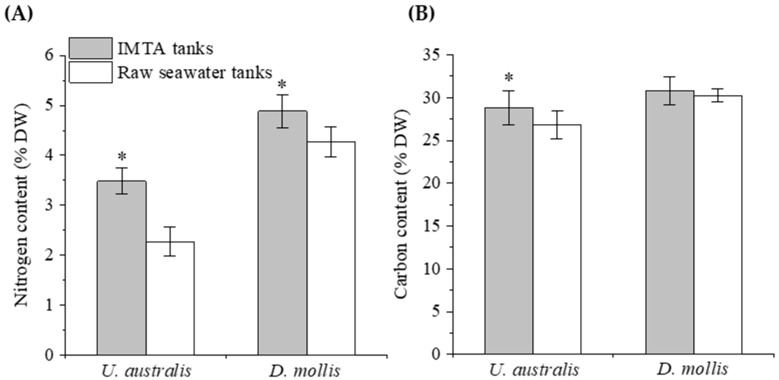
Nitrogen (**A**) and carbon (**B**) content of *U. australis* and *D. mollis* cultured in the IMTA and raw seawater system, respectively. Asterisks (*) indicate significant difference between treatments (*p* < 0.05).

**Figure 2 biology-14-01294-f002:**
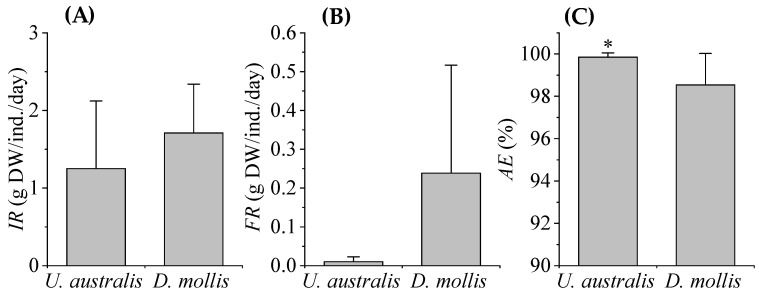
The ingestion rate (IR) (**A**), fecal rate (FR) (**B**) and absorption efficiency (AE) (**C**) of red sea urchins fed with *U. australis* and *D. mollis* cultured with the effluent of WSB from the IMTA system. Asterisks (*) indicate significant difference between treatments (*p* < 0.05).

**Figure 3 biology-14-01294-f003:**
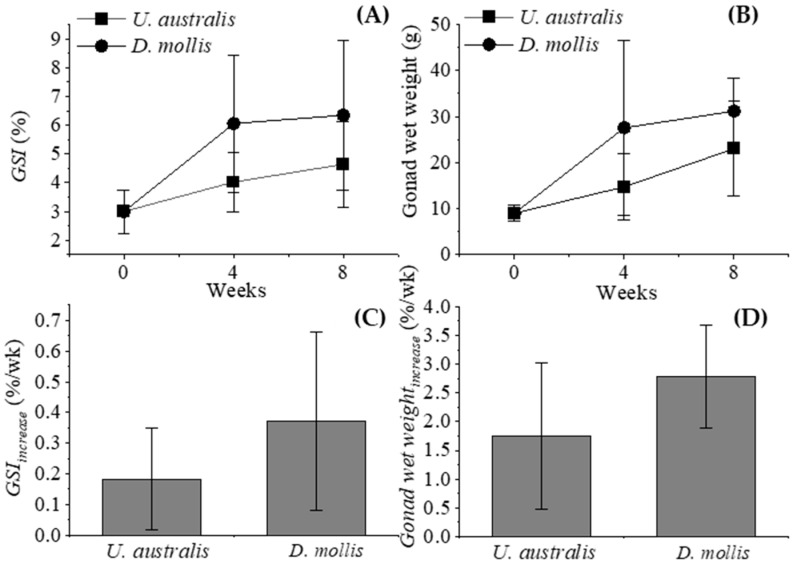
Gonadosomatic index (GSI) (**A**), gonad wet weight (WW) (**B**), average GSI_increase_ per week (**C**) and Gonad weight*_increase_* per week (**D**) of red sea urchins at time zero, four and eight weeks when fed IMTA-produced *U. australis* and *D. mollis* during the experimental period.

**Table 1 biology-14-01294-t001:** Criteria scale for evaluation of red sea urchin gonads quality and grade (Modified from [40]).

Color	Texture	Firmness	Grade
Level I—Bright yellow-orange	Fine	F	A
		NF	A
	Medium	F	A
		NF	A
	Coarse	F	B
		NF	B
Level II—Dull yellow-orange to light orange-brown	Fine	F	B
		NF	B
	Medium	F	B
		NF	B
	Coarse	F	C
		NF	C
Level III—Light to moderate brown	Fine	F	C
		NF	C
	Medium	F	C
		NF	C
	Coarse	F	D
		NF	D
Level IV—Dark brown or black	Fine	F	D
		NF	D
	Medium	F	D
		NF	D
	Coarse	F	D
		NF	D

Note: “A” grade = premium quality; “B” grade = high quality; “C” grade = mediocre quality; “D” grade = unacceptable quality; “F” = firm gonad; “NF” = not firm.

**Table 2 biology-14-01294-t002:** Color grading levels for red sea urchin gonads based on red, green and blue (RGB) values.

Level	Description	Color Display	RGB Value Range
Level I	Bright yellow-orange		R: 160–255, G: 115–255, B: 0–125
Level II	Dull yellow-orange to light orange-brown		R: 120–160, G: 75–125, B: 0–95
Level III	Light to moderate brown		R: 100–120, G: 50–115, B: 0–65
Level IV	Dark brown or black	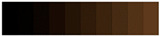	R: 0–100, G: 0–80, B: 0–35

**Table 3 biology-14-01294-t003:** Red sea urchin (RSU) gonad quality at the start of the trial (**A**), after four weeks of feeding with *U. australis* (**B**) and *D. mollis* (**C**) and after eight weeks of feeding with *U. australis* (**D**) and *D. mollis* (**E**).

**(A)**				
**RSU**	**Color Level**	**Texture (mm)**	**Firmness**	**Grade**
1	IV	0.68	F	D
2	III	1.23	NF	C
3	III	1.15	NF	C
4	IV	1.25	F	D
5	IV	0.98	F	D
**(B)**				
**RSU**	**Color Level**	**Texture (mm)**	**Firmness**	**Grade**
1	II	1.17	NF	B
2	IV	1.59	F	D
3	III	0.68	F	C
4	IV	0.81	F	D
5	II	0.87	NF	B
**(C)**				
**RSU**	**Color Level**	**Texture (mm)**	**Firmness**	**Grade**
1	II	0.75	F	B
2	II	1.14	F	B
3	I	0.92	F	A
4	III	0.78	NF	C
5	II	0.91	F	B
**(D)**				
**RSU**	**Color Level**	**Texture (mm)**	**Firmness**	**Grade**
6	II	0.77	F	B
7	II	0.91	F	B
8	II	0.88	F	B
9	II	0.62	F	B
10	II	1.03	F	B
**(E)**				
**RSU**	**Color Level**	**Texture (mm)**	**Firmness**	**Grade**
6	II	0.79	F	B
7	II	1.04	F	B
8	II	1.04	F	B
9	III	1.09	NF	C
10	II	1	F	B

## Data Availability

The original contributions presented in the study are included in the article. If there is a request, further inquiries can be directed to the corresponding author.

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
