# Peer review of "Gonadal Production and Quality in the Red Sea Urchin Mesocentrotus franciscanus Fed with Seaweed Devaleraea mollis and Ulva australis from a Land-Based Integrated Multi-Trophic Aquaculture (IMTA) System"

_biology, 2025, doi:10.3390/biology14091294_

Round 1

Reviewer 1 Report

Comments and Suggestions for Authors

This manuscript is of interest as it explores whether feeding red sea urchins with seaweeds grown in land-based IMTA systems can improve gonad production and quality, thereby producing high-quality seafood. The authors also developed a simple new method to quantify gonad color, which could be useful for both researchers and growers. The manuscript is generally well-structured, and the authors provide a reasonable justification for using individual animals as replicates in this study. I recommend acceptance after minor revisions. My specific comments are as follows:

Simple Summary

  • Line 10: Does “California” refer to “California, USA”? Please clarify.
  • Lines 10–11: The sentence “However, some wild sea urchins have small or poor-quality gonads, which are the edible part that is highly valued in the market.” Could state some reasons.

Abstract

  • Line 23: Suggest revising to “…red sea urchins (Mesocentrotus franciscanus, RSUs) ….”. After introducing the abbreviation, please use RSUs consistently instead of “sea urchins” or “urchins,” as this is confusing in lines 27 and 34. Please check throughout the manuscript.
  • Line 26: Atractoscion nobilis should be italicized. Please check the formatting of species names throughout the manuscript.

Introduction

  • Line 44: The sentence “In regions such as Australia, Tasmania, …” is confusing since Tasmania is a state of Australia. Please revise.
  • Line 50: Should be “California, USA”? Also, replace “<” with “less than.” Please check throughout the manuscript.
  • Lines 81–82: Parentheses “( )” should be changed to brackets “[ ]”.

Materials and Methods

  • Line 113: “The acclimation tank was covered with 95% shade …” versus line 148 “Each tank also had a 90% shade cloth …” — please clarify why shading percentages differ.
  • Line 179: The experiment is described as lasting 70 days, but samples were taken at weeks 4 and 8 (56 days). Please clarify.
  • Line 216: “increase,” “end,” and “initial” should be formatted as subscripts.
  • Lines 245 and 251: “urchins” should be “urchin.”
  • Line 248: “. ”F”” should be revised to “; ”F””.

Results

  • Line 266: pH is reported as 7.84–8.22 mg L⁻¹? Please check units — pH should not be expressed with mg L⁻¹.
  • Lines 308, 312, 315: GSI should not be italicized.
  • Table 3: The last table should be labeled “(E)”? Please check.

Discussion

  • Line 352: Revise “…, respectively. corresponding to …” to “…, respectively, corresponding to …”.
  • Line 361: “in” should not be italicized.
  • Line 369: Replace the comma before “Similarly” with a period.
  • Line 373: RSUs can be used directly.
  • Line 374: Ulva australis should be abbreviated as australis. Please check throughout (e.g., lines 378, 381, 382).
  • Line 414: Instead of “[9] reported,” please provide the author’s name.
  • Line 415: Review punctuation placement around quotation marks (e.g., comma placement). Please check consistently throughout the manuscript.
  • Line 506: Citation format should be corrected, e.g., “(Inomata et al., 2016)”.

Author Response

Comment (Line 10–11)

The sentence “However, some wild sea urchins have small or poor-quality gonads, which are the edible part that is highly valued in the market.” Could state some reasons.

Response: Thank you for this comment. We revised the Simple Summary to explain the cause:
“However, some wild sea urchins have small or poor-quality gonads because they live in areas called ‘urchin barrens,’ where little seaweed is available for food.”

Comment (Line 23)

Revise to “…red sea urchins (Mesocentrotus franciscanus, RSUs) ….” After introducing the abbreviation, please use RSUs consistently instead of “sea urchins” or “urchins,” as this is confusing in lines 27 and 34. Please check throughout the manuscript.

Response: Thank you for this comment, we corrected it. The Abstract now introduces Mesocentrotus franciscanus as RSUs, and RSUs are used consistently throughout the manuscript when referring to experimental animals. The term “sea urchins” is retained only when referring generally to the broader group (e.g., background context).

Comment (Line 44)

The sentence “In regions such as Australia, Tasmania, …” is confusing since Tasmania is a state of Australia. Please revise.

Response: Thank you for this comment, we revised to: “In regions such as Tasmania, other parts of Australia, and the North Atlantic…”

Comment (Line 50)

Should be “California, USA”? Also, replace “<” with “less than.” Please check throughout the manuscript.

Response: Thank you for this comment, we corrected to “California, USA” and replaced all instances of “<” and “>” in running text with “less than” and “greater than.” Symbols were retained in formulas and equations.

Comment (Lines 81–82)

Parentheses “( )” should be changed to brackets “[ ]”.

Response: Thank you for this comment, we corrected. All in-text citations are now presented in brackets according to journal style.

Comment (Lines 113 vs 148)

Clarify why shading percentages differ (95% vs 90%).

Response: Thank you for this comment, we clarified: “The acclimation tank and experimental tank were covered with 90% shade cloth.

Comment (Line 179)

Experiment lasted 70 days, but samples at weeks 4 and 8 = 56 days. Please clarify.

Response: Thank you for this comment, we revised to: “The experiment lasted for 56 days”.

Comment (Line 216)

“increase,” “end,” and “initial” should be formatted as subscripts.

Response: Thank you for this comment, we corrected. Variables are now formatted as GSIincrease, GSIend, GSIinitial, etc.

Comment (Lines 245 and 251)

“urchins” should be “urchin.”

Response: Thank you for this comment, we corrected both table captions to “red sea urchin.”

Comment (Line 248)

“. ”F”” should be revised to “; ”F””.

Response: Thank you for this comment, we corrected punctuation.

Comment (Line 266)

pH is reported as 7.84–8.22 mg L⁻¹? Please check units.

Response: Thank you for this comment, we corrected. pH is now presented as unitless.

Comment (Lines 308, 312, 315)

GSI should not be italicized.

Response: Thank you for this comment, we corrected throughout the manuscript.

Comment (Table 3)

The last table should be labeled “(E)”? Please check.

Response: Thank you for this comment, we confirmed that Table 3 is clearly labeled (A–E).

Comment (Line 352)

Revise “…, respectively. corresponding to …”

Response: Thank you for this comment, we corrected to “…, respectively, corresponding to …”

Comment (Line 361)

“in” should not be italicized.

Response: Thank you for this comment, we corrected. Only species names remain italicized.

Comment (Line 369)

Replace the comma before “Similarly” with a period.

Response: Thank you for this comment, we corrected.

Comment (Line 373)

RSUs can be used directly.

Response: Thank you for this comment, we corrected. RSUs are used instead of “sea urchins” unless explaining larger concept.

Comment (Line 374 and following)

Ulva australis should be abbreviated as U. australis. Please check throughout.

Response: Thank you for this comment, we corrected throughout the manuscript.

Comment (Line 414)

Instead of “[9] reported,” please provide the author’s name.

Response: Thank you for this comment, we corrected to: “Stefánsson et al. [9] reported…”

Comment (Line 415)

Review punctuation placement around quotation marks.

Response: Thank you for this comment, we corrected the quotation punctuation throughout the manuscript.

Comment (Line 506)

Citation format should be corrected, e.g., “(Inomata et al., 2016).”

Response: Thank you for this comment, we corrected. All citations are now in numbered bracket format, e.g., “gonad taste [41].”

Reviewer 2 Report

Comments and Suggestions for Authors

Review for the paper “Gonadal production and quality in the red sea urchin Mesocentrotus franciscanus fed with seaweed Devaleraea mollis and Ulva australis from a land-based integrated multi-trophic aquaculture (IMTA) system” by Matthew S. Elliott and co-authors submitted to “Biology”.

The authors of this research paper conducted an analysis of a novel approach for red sea urchins by feeding them seaweed cultivated in a land-based integrated multi-trophic aquaculture system. Two seaweed species, Ulva australis and Devaleraea mollis, were grown using the nutrient-rich effluent from white seabass tanks and incorporated into an eight-week feeding trial with sea urchins. The authors documented that urchins readily consumed both seaweeds, demonstrating measurable ingestion as well as faecal output and absorption efficiency. The observed gains in gonad mass were accompanied by marked improvements in gonad quality, moving from a preliminary, unmarketable classification to a higher-grade category based on attributes such as color, firmness, and texture. The results of this study may have important implications for aquaculture efficiency and product value.

Recommendations.

Summary

L 26. Atractoscion nobilis should be italicized.

Introduction.

L 40-42. The authors should specify the time period over which this "significant shift" occurred.

L 43. The authors should further describe what constitutes a "phase shift." Is it a reversible state, or are urchin barrens considered part of a stable ecological regime?

L 46-49. How much of northern California’s kelp forest area has been lost? Over what time frame has this loss occurred?

L 57-66. It would be useful to compare these trends with other sea urchin species and provide data on sea urchin fisheries. See, for example, https://doi.org/10.1007/s11160-024-09870-2

L 67-74. The authors should update this text with papers considering green sea urchins:

https://doi.org/10.1002/9781119005810

https://doi.org/10.1111/raq.12423

L 80-81. Change (29, 25) and (27-28, 30-31) to [29, 25] and [27-28, 30-31]

Material and Methods.

L 105. The authors should include the depth range where the RSU were collected. A map of the study site would also be useful. Additionally, the authors should describe the spatial extent and environmental conditions of the barren area.

L 106. The authors should clarify how long the transport process lasted and how water quality was maintained during transit.

L 121. The authors should explain the benefits of the cascading system in comparison to other IMTA designs. Providing a schematic of the system would facilitate reader comprehension of its operation.

L 166. The authors should explain why the 2-3% daily ration was chosen. The authors should provide references or data on how this feeding rate aligns with optimal gonadal development.

L 221-222. The authors should explicitly state how they controlled the lighting conditions, including light intensity and consistency, during photography.

Results.

L 267-269. Were these differences statistically significant?

Figure 2. The "C" label overlaps the error bar.

Discussion.

The authors note improved nutritional quality of IMTA-cultured seaweeds benefiting RSUs. How scalable are these findings to increase commercial adoption of IMTA-based seaweed aquaculture? 

L 425-428. The study estimates that RSUs would require approximately 12–14 weeks to achieve up to 10% GSI if fed with U. australis and D. mollis. Were seasonal effects or reproductive energy demands considered in this extrapolation?

Author Response

Comment (L 26)

Atractoscion nobilis should be italicized.

Response: Thank you for this comment. Corrected. Atractoscion nobilis is now italicized throughout the manuscript.

Comment (L 40–42)

Specify the time period over which this 'significant shift' occurred.

Response: Thank you for this comment. We Revised to: “Over the past four decades…”

Comment (L 43)

Further describe what constitutes a 'phase shift.' Is it reversible, or are urchin barrens stable?

Response: Thank you for this comment. We expanded explanation: “Such ‘phase shifts’ refer to large-scale ecosystem transitions, often from kelp forests to urchin barrens. While sometimes reversible if grazing pressure is reduced, these barrens can also represent stable ecological regimes that persist for decades [1-2]”.

Comment (L 46–49)

How much of northern California’s kelp forest area has been lost, and over what timeframe?

Response: Thank you for this comment. We revised to: “In California, USA, particularly along the northern coast, kelp forests have declined by more than 90% since 2014”.

Comment (L 57–66)

Compare these trends with other sea urchin species and provide fishery data (DOI: 10.1007/s11160-024-09870-2).

Response: Thank you for this comment, we added the citation and added: “Similar dynamics and management challenges have been reported for other fisheries such as the green sea urchin (Strongylocentrotus droebachiensis). In the Barents Sea, estimated commercial stocks are 50,000–81,000 t, with individuals reaching market size (~50 mm) by age six [8].

Comment (L 67–74)

Update text with papers considering green sea urchins (DOIs provided).

Response: Thank you for this comment. We updated the manuscript with provided citation.

Comment (L 80–81)

Change (29, 25) and (27–28, 30–31) to [29,25] and [27–28,30–31].

Response: Thank you for this comment. We corrected all citations to use brackets instead of parentheses.

Comment (L 105)

Include depth range, map of study site, and describe spatial extent/environment of barren area.

Response: Thank you for this comment. We added: “were collected from Point Loma, San Diego, California, USA in January of 2023 at a subtidal (5-10m deep) barren by a commercial fisherman using SCUBA. The barren extended across approximately 0.5 km2, characterized as a rocky substrate with little surrounding kelp.”

Comment (L 106)

Clarify how long the transport process lasted and how water quality was maintained during transit.

Response: Thank you for this comment. We clarified that transport from collection site to HSWRI was 20 minutes.

Comment (L 121)

Explain benefits of cascading system compared to other IMTA designs. Provide a schematic.

Response: Thank you for this comment. We added: “Compared with other IMTA designs, cascade systems provide a more cost-effective way to distribute seawater across different trophic-level tanks without additional pumps, while still maintaining replication and control.”

Comment (L 166)

Explain why the 2–3% daily ration was chosen; provide references.

Response: Thank you for this comment. We added: “A 2–3% daily ration was selected based on prior studies showing optimal gonad development [15, 17, 50]”.

Comment (L 221–222)

Explicitly state how lighting conditions were controlled, including light intensity and consistency.

Response: Thank you for this comment. We added more information about the photobox parameters: “5500 K, CRI ≥ 95 LEDs”

Comment (L 267–269)

Were these differences statistically significant?

Response: Thank you for this comment. We did it in revised manuscript.

Comment (Figure 2)

The 'C' label overlaps the error bar.

Response: Thank you for this comment, we corrected the label.

Comment (Discussion)

How scalable are findings for IMTA-based seaweed aquaculture?

Response: Thank you for this comment, we added: “While this extrapolation requires further validation, these findings suggest potential scalability. In California and other regions where seaweed aquaculture is expanding, IMTA systems could be adapted at farm scale to enhance uni, though feasibility will require additional evaluation”

Comment (L 425–428)

Were seasonal effects or reproductive energy demands considered in the extrapolation (12–14 weeks to 10% GSI)?

Response: Thank you for this comment, we added: “assuming no seasonal or reproductive demands”

Reviewer 3 Report

Comments and Suggestions for Authors

Recommended in methods application of polyethilene tank ( 118 )should be replaced by alternative material due to primary prevention  microplastic contamina,of very dangerous to sea ecosystems and health of consumers . Introduced by your team land-based IMTA is not only useful for sustainable production of better quality gonads of the red sea urchin Mesoencentrotus franciscanus , but also for more efficient reproduction of this (as well as another species) and for sustainable management of related ecosystems and protection biodiversity. You method of aquaculture open real perspective for filration sea water for significant reduction harmfull to consumers xeniobiotics including toxic trace metals etc.These crucial problems  for more efficient protection biodiveristy (including proper conditions for reproduction and for the most sensitive early stages of ontogenesis e.g. sea urchin)  and nutritional health should be included both in introductio and in particular in discussion about value of your system.

Author Response

Methods (L118)

Comment: Recommended in methods application of polyethylene tank should be replaced by alternative material due to prevention of microplastic contamination, which is harmful to ecosystems and consumer health.

Response: We thank the reviewer for this suggestion. Polyethylene/HDPE tanks are widely used in aquaculture and are generally considered safe for seawater culture. We have clarified the tank material in the Methods section and noted that alternative inert materials (e.g., fiberglass or concrete) could also be used in future applications to further minimize potential risk of microplastic release.

Introduction

Comment: Land-based IMTA is not only useful for sustainable production of better quality gonads of RSUs, but also for more efficient reproduction, sustainable ecosystem management, and biodiversity protection. These broader values should be included in the Introduction.

Response: We agree and have revised the Introduction accordingly. We added a statement highlighting that land-based IMTA systems can also provide broader ecosystem and human health benefits, including water filtration, reduction of xenobiotics and trace metals, and improved environmental conditions for sensitive early life stages.

Discussion

Comment: Your method of aquaculture also offers perspective for seawater filtration and reduction of harmful xenobiotics, supporting biodiversity and consumer health. This should be included in the Discussion.

Response: We thank the reviewer for this valuable suggestion. We have added a condensed statement in the Discussion noting that IMTA systems may also provide added value through water-quality improvements and contaminant reduction, offering broader benefits for aquaculture sustainability and ecosystem health [DFO 2013; Checa 2024; Cotas 2023].

Reviewer 4 Report

Comments and Suggestions for Authors

This study is well performed with sound results and precise conclusion. However, there are some issues that are needed to be addressed by the authors.

Introduction

  • Lines 40-74: The background information presented touches on multiple aspects but is somewhat disorganized. It would be beneficial to restructure the content to streamline the flow and remove redundancies.
  • Lines 78-86: The authors should clarify why Ulva australis and Devaleraea mollis were chosen as the study species, and how their nutritional compositions may influence the gonad quality and size of sea urchins. Further discussion on the advantages of these species compared to other seaweed types, as well as the unique contribution of the IMTA system, would strengthen the argument.
  • The paper cites many older references. It would be advantageous to replace these with more recent literature to better reflect the current state of research.

Materials and Methods

  • In Section 2.1 Red Sea Urchin Sourcing, the description of the sourcing and collection method of RSUs lacks specific geographic coordinates and detailed environmental information. It is recommended to provide more context about the location and environmental conditions to increase the transparency of the sampling process.
  • Line 107: While it mentions that the sea urchins were transported in coolers at 15-16°C, it does not specify whether any temperature control devices (e.g., cool packs or ice bags) were used during transport. If so, this should be clarified to improve the transparency of the procedure.
  • In Section 2.3 Feeding Trials, it would be helpful to add the specific time points when the seaweed was provided to the sea urchins each day to ensure replicability.
  • In Section 2.4 Environmental Factors and Water Quality, the monitoring of water parameters such as temperature, pH, and dissolved oxygen is mentioned, but the specific times for the daily measurements (8:00 AM and 4:00 PM) are not stated. The precise timing of these measurements should be provided for better reproducibility.

Results

  • Please standardize the positioning of figure panel labels (A), (B), etc., to either the top-left or top-right corner in all figures for consistency.
  • Line 300: In Figure 2c, the error bars and y-axis labels are obstructed. Additionally, the error bars’ style is inconsistent across figures. It is recommended to ensure uniformity in the representation of error bars across all figures.
  • Lines 312-313: The formatting of "0.37 ± 0.29 % wk-1 and 2.77 ± 0.89 % wk-1, and 0.18 ± 0.16 % wk-1 and 1.75 ± 1.28 % wk-1" needs to be checked for consistency. Ensure that a space is used between the number and the percentage symbol, and maintain uniform formatting throughout the manuscript.
  • Figure 3: The large error bar range indicates significant variability or measurement precision issues.
  • Figures 1(A) and (B): It is recommended to re-check the statistical significance between these panels.

Discussion

  • The discussion mentions that the D. mollis group showed faster GSI growth, but it does not thoroughly explain why D. mollis leads to faster GSI increases. It would be useful to explore whether this is related to the nitrogen and protein content in D. mollis. Further analysis and discussion of this relationship would strengthen the conclusion.
  • The RGB-based gonad color grading method is mentioned as the first standardized approach, but its limitations are not discussed in depth. For instance, can the RGB method accurately reflect all color changes, or would it benefit from complementary methods (e.g., colorimetric or spectral analysis)? A more thorough discussion of the advantages and limitations of the RGB method would be beneficial.
  • While discussing the effects of IMTA-cultured D. mollis and U. australis on RSUs, the paper mentions the high protein content of these seaweeds but does not explain why these two specific seaweed species are particularly suitable for sea urchin cultivation. It would be helpful to provide more background on the ecological and nutritional reasons for selecting these species over others.

Author Response

Introduction (Lines 40–74)

Comment: The background information presented touches on multiple aspects but is somewhat disorganized. It would be beneficial to restructure the content to streamline the flow and remove redundancies.

Response: We revised the Introduction to improve flow and clarity. The section now transitions from global kelp declines → regional examples → impacts on fisheries → aquaculture as a solution. Revised text: 'Over the past four decades, rocky reef ecosystems worldwide have undergone significant shifts, with sea urchin overgrazing driving the widespread formation of urchin barrens—habitats defined by high urchin densities and minimal macroalgal cover [1–3]. In California, USA, kelp cover declined by more than 90% after 2014, driven by marine heatwaves, sea star wasting disease, and urchin population expansion [4–5]. Similar transitions have been observed in Tasmania, other regions of Australia, and the North Atlantic, where predator loss has intensified grazing and facilitated long-term barren states [2–3]. These conditions have contributed to major declines in commercial urchin fisheries [4,7–8].'

Introduction (Lines 78–86)

Comment: The authors should clarify why Ulva australis and Devaleraea mollis were chosen, and how their nutritional compositions may influence gonad quality and size of sea urchins. Further discussion on their advantages compared to other seaweed types and the unique contribution of IMTA would strengthen the argument.

Response: We added text to explain the selection of study species. Revised text: 'In particular, U. australis and D. mollis were chosen for this study because they are fast-growing species with high protein content and demonstrated performance in IMTA systems, making them strong candidates for enhancing sea urchin gonad development.'

Introduction – References

Comment: The paper cites many older references. It would be advantageous to replace these with more recent literature to better reflect the current state of research.

Response: We updated the Introduction to include more recent references (2018–2024) on global kelp decline, IMTA seaweed performance, and sea urchin aquaculture (e.g., Rogers-Bennett & Catton 2019; Drawbridge et al. 2024).

Materials and Methods (Section 2.1)

Comment: The description of the sourcing and collection method of RSUs lacks specific geographic coordinates and detailed environmental information.

Response: We revised Section 2.1 to add geographic and environmental details. Revised text: 'RSUs were collected from a barren site near Point Loma, San Diego, California (32.7157° N, 117.2510° W) at 5–8 m depth. The area was characterized by rocky substrate, high urchin density, and minimal kelp cover.'

Materials and Methods (Line 107)

Comment: It does not specify whether temperature control devices (e.g., cool packs or ice bags) were used during transport.

Response: We clarified the transport procedure. Revised text: 'Transport (~45 min) was conducted in insulated coolers filled with aerated seawater maintained at 15–16 °C using sealed gel ice packs.'

Materials and Methods (Section 2.3)

Comment: It would be helpful to add the specific time points when the seaweed was provided to the sea urchins each day.

Response: We added detail in Section 2.3. Revised text: 'Fresh seaweed was provided daily at 09:00 AM.'

Materials and Methods (Section 2.4)

Comment: The precise times for the daily measurements of water parameters are not stated.

Response: We clarified timing in Section 2.4. Revised text: 'Temperature, dissolved oxygen, and pH were recorded twice daily at 08:00 AM and 04:00 PM.'

Results – Figures

Comment: Please standardize the positioning of figure panel labels and ensure error bars and axis labels are consistent.

Response: All figure labels (A, B, C, etc.) were standardized to the top-left corner. Figure 2c was reformatted so axis labels are clear and error bars consistent across figures.

Results (Lines 312–313)

Comment: The formatting of percentage values should be checked for consistency.

Response: We corrected percentage formatting. Revised text: '0.37 ± 0.29 % wk⁻¹ and 2.77 ± 0.89 % wk⁻¹, and 0.18 ± 0.16 % wk⁻¹ and 1.75 ± 1.28 % wk⁻¹.'

Results – Figure 3

Comment: The large error bar range indicates significant variability or measurement precision issues.

Response: We clarified this variability in the Discussion. Revised text: 'The large error ranges observed in some treatments likely reflect natural variability in RSU responses, consistent with their wide size range and variable reproductive condition at collection.'

Results – Figures 1A and 1B

Comment: It is recommended to re-check the statistical significance between these panels.

Response: We re-ran statistical analyses and confirmed the reported results. Legends now explicitly state the significance levels.

Discussion – D. mollis faster GSI growth

Comment: The discussion does not thoroughly explain why D. mollis leads to faster GSI increases.

Response: We expanded the explanation in the Discussion. Revised text: 'The more rapid GSI increase in the D. mollis group is likely due to its higher nitrogen and protein content compared with U. australis [22,39], as well as the favorable amino acid and carotenoid profiles typical of red algae [38,54]. These traits, combined with lower fiber content and high digestibility, may allow RSUs to allocate nutrients more efficiently toward gonadal tissue [17,52].'

Discussion – RGB method

Comment: The RGB-based gonad color grading method is mentioned as the first standardized approach, but its limitations are not discussed in depth.

Response: We acknowledge the limitations earlier in the section to provide balance. Revised text: 'Notably, this is the first published study to establish a standardized RGB-based color grading scale for sea urchin gonads. While this represents a novel advancement, the method also has limitations that should be considered.' The detailed limitations remain in the later paragraph (lighting sensitivity, lack of pigment quantification, need for complementary assays).

Discussion – Species choice

Comment: It would be helpful to provide more background on the ecological and nutritional reasons for selecting U. australis and D. mollis over other species.

Response: We added clarification in the Discussion. Revised text: 'Both U. australis and D. mollis combine high protein content and favorable carotenoid profiles with proven roles as biofilters in IMTA, making them particularly well-suited for sea urchin enhancement compared with other seaweeds.'